# Pitavastatin and Ivermectin Enhance the Efficacy of Paclitaxel in Chemoresistant High-Grade Serous Carcinoma

**DOI:** 10.3390/cancers14184357

**Published:** 2022-09-07

**Authors:** Mariana Nunes, Diana Duarte, Nuno Vale, Sara Ricardo

**Affiliations:** 1Differentiation and Cancer Group, Institute for Research and Innovation in Health (i3S) of the University of Porto/Institute of Molecular Pathology and Immunology of the University of Porto (IPATIMUP), 4200-135 Porto, Portugal; 2Institute of Biomedical Sciences Abel Salazar (ICBAS), University of Porto, 4050-313 Porto, Portugal; 3OncoPharma Research Group, Center for Health Technology and Services Research (CINTESIS), 4200-450 Porto, Portugal; 4CINTESIS@RISE, Faculty of Medicine, University of Porto, 4200-319 Porto, Portugal; 5Department of Community Medicine, Health Information and Decision (MEDCIDS), Faculty of Medicine, University of Porto, 4200-319 Porto, Portugal; 6Toxicology Research Unit (TOXRUN), University Institute of Health Sciences, Polytechnic and University Cooperative (CESPU), CRL, 4585-116 Gandra, Portugal; 7Department of Pathology, Faculty of Medicine, University of Porto (FMUP), 4200-319 Porto, Portugal

**Keywords:** chemoresistance, drug repurposing, paclitaxel, high-grade serous carcinoma, combinatory regiment, synergy

## Abstract

**Simple Summary:**

The main challenge in high-grade serous carcinoma management is to unveil therapeutic approaches to overcome chemoresistance. Drug combinations and repurposing of non-oncological agents are attractive strategies that allow for higher efficacy, decreased toxicity, and the overcoming of chemoresistance. Several non-oncological drugs display an effective anti-cancer activity and have been studied to be repurposed in multi-drug resistant neoplasms. The purpose of our study was to explore whether combining Paclitaxel with repurposed drugs (Pitavastatin, Metformin, Ivermectin, Itraconazole and Alendronate) led to a therapeutic benefit. Our results showed that the combination of Paclitaxel with Pitavastatin or Ivermectin demonstrates the highest cytotoxic effect and the strongest synergism among all combinations for two chemoresistant cell lines. Thus, the combination of these repurposed drugs with Paclitaxel could be a particularly valuable strategy to treat ovarian cancer patients with intrinsic or acquired chemoresistance.

**Abstract:**

Chemotherapy is a hallmark in high-grade serous carcinoma management; however, chemoresistance and side effects lead to therapeutic interruption. Combining repurposed drugs with chemotherapy has the potential to improve antineoplastic efficacy, since drugs can have independent mechanisms of action and suppress different pathways simultaneously. This study aimed to explore whether the combination of Paclitaxel with repurposed drugs led to a therapeutic benefit. Thus, we evaluated the cytotoxic effects of Paclitaxel alone and in combination with several repurposed drugs (Pitavastatin, Metformin, Ivermectin, Itraconazole and Alendronate) in two tumor chemoresistant (OVCAR8 and OVCAR8 PTX R P) and a non-tumoral (HOSE6.3) cell lines. Cellular viability was assessed using Presto Blue assay, and the synergistic interactions were evaluated using Chou–Talalay, Bliss Independence and Highest Single Agent reference models. The combination of Paclitaxel with Pitavastatin or Ivermectin showed the highest cytotoxic effect and the strongest synergism among all combinations for both chemoresistant cell lines, resulting in a chemotherapeutic effect superior to both drugs alone. Almost all the repurposed drugs in combination with Paclitaxel presented a safe pharmacological profile in non-tumoral cells. Overall, we suggest that Pitavastatin and Ivermectin could act synergistically in combination with Paclitaxel, being promising two-drug combinations for high-grade serous carcinoma management.

## 1. Introduction

Cytoreductive surgery and chemotherapy remain the most common therapeutical options in high-grade serous carcinoma (HGSC) management; however, many patients still experience recurrence, manifesting by the presence of malignant ascites and characterized by intratumoral heterogeneity and resistance to conventional antineoplastic agents [1,2,3]. Paclitaxel is a mitotic inhibitor employed in the treatment of many types of cancers, including advanced HGSC [4,5] that targets β-tubulin, a protein responsible for stabilizing the microtubule polymers, blocking cells in phases G0/G1 and G2/M, resulting in tumor cell death [6]. However, Paclitaxel treatment is limited by the acquired chemoresistance and its severe side effects, such as peripheral neuropathy, leading to the chemotherapy interruption [7,8]. Several molecular factors may contribute to acquired chemoresistance in HGSC such as modification of drug targets, decreased cellular drug accumulation, increased expression of drug pumps and/or detoxification systems, improved DNA repair process and, reduced sensitivity to apoptosis and enhanced proliferation [9,10,11]. Current strategies to delay chemoresistance and achieve a significant therapeutic outcome consist in the development of new synthetic analogues of existing drugs, targeted therapy, immunotherapy, sequential strategies, combinatory regiments, among others [9,12]. Some of these therapeutic strategies, such as the development of new drugs, implicate a time- and cost-consuming process, accompanied by a low approval rate and occasionally with severe side effects [13]. Thus, it is crucial to explore more advantageous pharmacological strategies to improve therapeutic regimens.

Drug repurposing is an approach to identify new purposes for existing drugs, already approved for other diseases [14,15,16]. This strategy is based on the fact that different diseases can share same therapeutic targets and molecular features, and in the concept of pleiotropic drug effect (i.e., a drug can have different effects than those for which was specifically developed) [17]. Drug repurposing allows for a more easily accessible alternative since all the developmental information concerning pharmacokinetics, pharmacodynamics, safety and toxicological profiles, dosage, drug interactions, mechanisms of action, molecular targets, and clinical experience is already well known, facilitating their introduction in clinical practice [18,19,20]. This understanding simplifies all the regulatory processes of drug approval and increases the success of clinical applicability, considering the therapeutic challenges of malignant diseases and the economic impact on health systems [21]. Nevertheless, drug repurposing implies the conducting of clinical trials to evaluate drug efficacy and estimate the maximum tolerated dose to avoid unacceptable toxicities [18].

A continuous single-agent treatment motivates tumor cells to search for alternative pathways, resulting in chemoresistance [22]. Combining two or more drugs that target different pathways increases sensitivity and treatment efficacy, becoming less prone to acquired chemoresistance and toxicity effects since lower therapeutic doses can be used [22,23,24,25,26] also to overcome intratumoral and intertumoral heterogeneity [27]. Since HGSC has a heterogeneous cancer cell population, it is essential to find alternative multitargeting drugs and combine them to create personalized therapies, increasing the chance of eradicating all cancer cells. The interaction between two or more drugs can be synergistic, additive, or antagonistic, being synergism the most desirable drug interaction in the pharmacological context. In a synergic interaction, the effect of combining drugs is much higher than the expected additive effect of each agent. In an additive interaction, the combined effect refers to the sum of the effects of each agent, while in the antagonism interaction, the combined effect is less potent than the sum of the single activity of each drug [28]. Therefore, when two drugs act synergistically, the possibility to achieve the desired outcome can be obtained using lower doses of each drug, reducing systemic toxicity, and minimizing the adverse side effects [23,29,30].

Several studies suggest that the antitumoral activity of the antineoplastic agents can effectively be improved by the combination with repurposed drugs, which have acceptable toxicological profile and can simultaneously increase the activity of the referenced dug and reduce therapeutical doses [31,32,33,34,35]. In our previous review, and based on several experimental and observational studies, we support the assumption that Pitavastatin, Metformin, Ivermectin, Itraconazole and Alendronate exhibit anti-tumoral effects in many tumoral contexts and have the potential of being used in the treatment of ovarian cancer (OC) [15]. We hypothesized that five repurposed drugs (Pitavastatin, Metformin, Ivermectin, Itraconazole and Alendronate) could act synergistically with Paclitaxel in two chemoresistant HGSC cell lines, e.g., OVCAR8 and OVCAR8 PTX R P cells. The OVCAR8 PTX R P is a Carboplatin and Paclitaxel-resistant cell line previously established in our laboratory from parental OVCAR8 (acquired Carboplatin-resistant cell line) by pulse exposure to a stepwise increasing PTX concentration [36]. Moreover, we will test these conditions in a non-tumoral cell lines (HOSE6.3) to achieve a safety pharmacological profile.

Here, we have demonstrated that Pitavastatin and Ivermectin have anticancer activity, being the most promising repurposed drugs when used alone, with the lowest half-maximal inhibitory concentration (IC_50_) value for both chemoresistant HGSC cell lines. The simultaneous combination of Paclitaxel with Pitavastatin or Ivermectin showed the highest cytotoxic effect and the strongest synergism for both HGSC chemoresistant cell lines, resulting in a chemotherapeutic effect superior to both drugs alone. Moreover, results for OVCAR8 PTX R P (Carboplatin and Paclitaxel-resistant) cells were even more promising than OVCAR8 (Carboplatin-resistant), with a higher number of synergistic pairs for all the combinations considered. Importantly, almost all the repurposed drugs tested alone and in combination with Paclitaxel presented a safety pharmacological profile, not demonstrating significant effects in the reduction of cellular viability of HOSE6.3, a non-tumoral cell line. These are promising results that may lead to new therapeutic strategies in HGSC management.

## 2. Materials and Methods

### 2.1. Cell Lines and Culture Conditions

OVCAR8 and OVCAR8 PTX R P cells were selected as HGSC models, particularly since they are described as Carboplatin-resistant [37] and Carboplatin and Paclitaxel-resistant [36] HGSC cell lines, respectively. OVCAR8 was kindly provided by Doctor Francis Jacob, Gynecological Cancer Center and Ovarian Cancer Research, Department of Biomedicine, University Hospital Basel and University of Basel, Basel, Switzerland. OVCAR8 PTX R P was previously established in our laboratory from parental OVCAR8 by pulse exposure [36]. Briefly, OVCAR8 PTX R P was maintained uninterruptedly in the presence of Paclitaxel for 1 month and after that were exposed to Paclitaxel for 4 days followed by recovery in drug-free media for 4 days (pulse-selection strategy), during another 2 months [36]. Additional experiments were carried out in a non-tumoral cell line (HOSE6.3), described as a human ovarian epithelial cell line established from a normal ovary, surgically removed from patients with non-malignant disease [38]. All cell lines were grown in complete media, i.e., RPMI-1640 medium, GlutaMAX^TM^ Supplement, HEPES (ThermoFisher Scientific, Waltham, MA, USA), supplemented with 10% (*v*/*v*) inactivated and filtered fetal bovine serum (FBS; Biowest, Nuaillé, France) and 1% (*v*/*v*) penicillin/streptomycin (ThermoFisher Scientific) and maintained at 37 °C and 5% CO_2_. Cells were authenticated using short tandem repeat profiling and regularly tested for the absence of mycoplasma.

### 2.2. Drugs

All drugs (i.e., Paclitaxel, Pitavastatin, Metformin, Ivermectin, Itraconazole and Alendronate) were purchased from Selleckchem (Houston, TX, USA), dissolved in dimethyl sulfoxide (DMSO; AppliChem, Barcelona, Spain) or distilled water and stored at −80°C, according to the manufacturer’s instructions. Immediately prior to use, an aliquot was diluted at the required concentrations.

### 2.3. Cell Viability Assay

To determine the effect of single and combination drug treatments on the cellular viability of cell lines, a resazurin-based assay—Presto Blue (PB)—was performed. Briefly, 5 × 10^3^ cells/well were seeded into a 96-well plate in complete media, incubated at 37 °C and 5% CO_2_ and allowed to adhere overnight prior to drug exposure. After 24 h, cells were exposed to increasing concentrations of drugs and incubated under the same conditions. After 48 h, culture medium was removed, and 50 μL/well of PrestoBlue^TM^ Cell Viability Reagent 1× (ThermoFisher Scientific) was added. Cells were incubated for 45 min, protected from light, at 37 °C and 5% CO_2_. Fluorescence was measured (560 nm excitation/590 nm emission) using a Bio Tek Synergy^TM^ 2 multi-mode microplate reader (BioTek, Winooski, VT, USA).

### 2.4. Drug Treatment

The IC_50_ values were first obtained for each drug alone for OVCAR8, OVCAR8 PTX R P and HOSE6.3 cell lines. The IC_50_ for all drugs was achieved by comparing treated cells with control cells (considered 100% viable) containing 1% (*v*/*v*) of the vehicle (DMSO or distilled water). For all experiments, no differences were observed between control cells with/without vehicle. For the single drug treatment, cells were treated for 48 h with Pitavastatin (0.04 to 5 μM) [39], Metformin (0.08 to 10 mM) [40], Ivermectin (0.39 to 50 μM) [41], Itraconazole (0.39 to 50 μM) [42], and Alendronate (7.81 to 1000 μM) [43]. Combination studies were performed according to the previously described method [31], using increasing concentrations of both drugs in a fixed ratio, as suggested by Chou–Talalay [44]. First, Paclitaxel (Drug 1) was combined in a simultaneous treatment with different repurposed drugs (Drug 2) in fixed-dose ratio that corresponds to 0.25, 0.5, 1, 2, and 4 times the individual IC_50_ values for 48 h.

### 2.5. Drug Interactions Analysis

To measure drug interaction between Paclitaxel (Drug 1) and repurposed drugs (Drug 2), we calculated the Combination Index (CI) by the Chou–Talalay method [45] using the CompuSyn Software (ComboSyn, Inc., New York, NY, USA). A mutually exclusive model, assuming that drugs act through entirely different mechanisms, was used for this analysis [46]. Drug 1 and Drug 2 were combined in a fixed ratio of doses that corresponded to 0.25, 0.5, 1, 2 and 4 times the individual IC_50_ values. The CI is a quantitative representation of pharmacological interactions (i.e., CI < 1, synergism; =1, additive interactions; and >1, antagonism), plotted on y-axis as a function of effect level (Fa) on the x-axis to assess drug synergism between drug combinations. Additionally, we estimated the expected drug combination responses based on the Bliss Independence and Highest Single Agent (HSA) reference models using SynergyFinder 2.0 Software [47]. Positive and negative synergy score values denote synergy and antagonism, respectively. The cNMF algorithm implemented in SynergyFinder 2.0 was used for estimation of outlier measurements [48].

### 2.6. Microscopic Evaluation

All microscopic figures were obtained under a Leica DMi1 inverted phase contrast microscope (Leica Microsystems, Wetzlar, Germany), at 50× magnification.

### 2.7. Statistical Analysis

All assays were performed in triplicate with at least three independent experiments. Data were expressed as mean ± standard deviation (SD), statistical analysis was carried out in GraphPad Prism 8 (GraphPad Software Inc., San Diego, CA, USA) using ordinary one-way or two-way ANOVA followed by Šıdák’s multiple comparison test.

## 3. Results

### 3.1. Repurposed Drugs Demonstrate High Efficacy in Reducing Cellular Viability of Chemoresistant HGSC Cells

First, we analyzed the anti-tumor potential of each repurposed drug as a single agent on OVCAR8 and OVCAR8 PTX R P cells using increasing concentrations of each drug (see Section 2.4) and evaluating the cellular viability after 48 h of treatment exposure. Our results could obtain a dose–response curve and could calculate the IC_50_ values that were further used in combinations studies. Herein, we have shown that all repurposed drugs displayed an anti-tumor activity in both chemoresistant HGSC cell lines (Figure 1). Pitavastatin showed an anti-cancer effect, being the strongest among all drugs tested alone and presenting the lowest IC_50_ values of 0.801 ± 0.061 and 0.850 ± 0.142 μM, for OVCAR8 and OVCAR8 PTX R P cells, respectively (Figure 1A). Cytotoxic effect of Metformin was achieved at high concentrations with IC_50_ values of 1.495 ± 0.169 and 1.458 ± 0.193 mM for OVCAR8 and OVCAR8 PTX R P cells, respectively (Figure 1B). Ivermectin exposure demonstrated a strong cytotoxic effect, revealing IC_50_ values of 15.960 ± 2.909 and 15.522 ± 1.859 µM for OVCAR8 and OVCAR PTX R P cells, respectively (Figure 1C). Cytotoxic effect of Itraconazole was attained at high concentrations with IC_50_ values of 39.118 ± 3.376 and 35.571 ± 0.480 µM for OVCAR8 and OVCAR PTX R P cells, respectively (Figure 1D). Alendronate treatment showed a cytotoxic effect with IC_50_ values of 219.068 ± 13.555 and 196.325 ± 20.234 µM for OVCAR8 and OVCAR8 PTX R P cells, respectively (Figure 1E). Additionally, to evaluate the effects of the repurposing drugs in normal-like cells, we tested increasing concentrations of each drug as a single agent on a human ovarian epithelial cell line (HOSE 6.3) described as a cell line obtained from a normal ovary, surgically removed from patients with non-malignant diseases [38]. Overall, our results demonstrate that the tested drugs had no or very low effect on the cellular viability of HOSE6.3 cells, contrary to the effects on tumoral cell lines (Appendix A). The cytotoxic effect of Pitavastatin, Metformin, and Alendronate showed a lack of efficacy in the reduction of cellular viability of HOSE6.3 cells and not obtaining IC_50_ values in the considered concentration ranges (Appendix A). Ivermectin and Itraconazole exposure demonstrated a cytotoxic effect for HOSE6.3 cells, at higher concentrations, revealing IC_50_ values of 12.635 ± 2.185 and 38.574 ± 5.112 µM, respectively (Appendix A). Regarding HOSE6.3 cells, and in agreement with the previously mentioned results, morphological differences were discreetly observed for the highest concentration tested for all the repurposing drugs compared to vehicle (Appendix A). These results confirmed that the five repurposed drugs have an acceptable safety profile in normal HOSE6.3 cells presenting simultaneously a significant anticancer efficacy in chemoresistant tumor cells, making them good candidates for being tested in combination with Paclitaxel.

### 3.2. Repurposed Drugs Increase the Efficacy of Paclitaxel in Reducing Cellular Viability of Chemoresistant HGSC Cells

After obtaining the IC_50_ values for all the drugs, we evaluated the combination of Paclitaxel with each repurposed drug using the combination model previously described [32]. Briefly, OVCAR8 and OVCAR8 PTX R P cells were exposed to two drugs (Drug 1 and Drug 2) alone and combined in a fixed ratio that corresponds to 0.25, 0.5, 1, 2, and 4 times the individual IC_50_ values of each drug (Figure 2 and Appendix A). Additionally, a morphological evaluation was performed for each treatment condition (Figure 3). For OVCAR8 cells, combining Paclitaxel with Pitavastatin resulted in a significant increase in anti-cancer effect of 0.25 (*p* < 0.005), 0.5 and 1 (*p* < 0.0001) times the IC_50_ values when compared to Paclitaxel alone (Figure 2A and Appendix A). Likewise, for OVCAR8 PTX R P cells, this combination produced a significant reduction in cellular viability (*p* < 0.0001) for all the concentrations than Paclitaxel as a single agent (Figure 2B and Appendix A). For OVCAR8 cells, the combination of Paclitaxel with Metformin caused a significant reduction in cellular viability for 0.25 (*p* < 0.001), 0.5 and 1 (*p* < 0.0001), and 2 (*p* < 0.05) times the IC_50_ values comparing to Paclitaxel alone (Figure 2C and Appendix A). Moreover, for OVCAR8 PTX R P cells, this combination demonstrated a significant increase in anti-tumor effect for all the concentrations tested (i.e., *p* < 0.001 for 0.25 and *p* < 0.0001 for 0.5, 1, 2 and 4 times the IC_50_ values) than Paclitaxel as a single agent (Figure 2D and Appendix A). For OVCAR8 cells, combining Paclitaxel with Ivermectin showed a significant increase in anti-cancer effect for all the tested concentrations (i.e., *p* < 0.05 for 0.25 and 4; and *p* < 0.0001 for 0.5, 1 and 2 times the IC_50_ values) compared to Paclitaxel as a single agent (Figure 2E and Appendix A). In addition, for OVCAR8 PTX R P cells, this combination resulted in a significant increase in anti-tumor effect (*p* < 0.0001) for 0.5, 1, 2 and 4 times the IC_50_ values, when compared to Paclitaxel (Figure 2F and Appendix A). For OVCAR8 cells, the combination of Paclitaxel with Itraconazole did not result in a significant reduction in cellular viability compared to Paclitaxel alone, demonstrating that this drug may be responsible for the observed combined effect (Figure 2G and Appendix A). However, this combination showed a significant increase in anti-cancer effect compared to Itraconazole as a single agent, for 0.5 (*p* < 0.005), 1, 2 and 4 (*p* < 0.0001) times the IC_50_ values (Figure 2G and Appendix A). For OVCAR8 PTX R P cells, this combination demonstrated a significant cellular viability reduction (*p* < 0.0001) for 1, 2 and 4 times the IC_50_ values than Paclitaxel alone (Figure 2H and Appendix A). For OVCAR8 cells, a significant decrease in cellular viability (*p* < 0.0001) between Paclitaxel with Alendronate and Paclitaxel alone was obtained at the concentration of one time the IC_50_ values (Figure 2I and Appendix A). Moreover, for OVCAR8 PTX R P cells, our results indicate a significant increase in anti-cancer effect for 0.5 (*p* < 0.005), and 1, 2 and 4 (*p* < 0.0001) times the IC_50_ values, concerning Paclitaxel alone (Figure 2J and Appendix A). Regarding OVCAR8 and OVCAR8 PTX R P cells, and in agreement with the previously mentioned results, morphological differences were observed for all the single and combined treatments compared to vehicle. The combinatory treatments induced a more aggressive phenotype, i.e., decreasing of cell number, less aggregate formation, and smaller and rounded cells, indicative of cell death compared to single treatments (Figure 3A,B). These phenotype results demonstrate the anti-cancer effect of all the repurposed drugs and support in the combinations tested in this study. For all the drug pairs tested, some concentration demonstrates a significant increase in anti-cancer effect when compared to both drugs alone. We showed better drug combination pairs for OVCAR8 PTX R P cells that are characterized by a double resistance profile (Carboplatin and Paclitaxel).

### 3.3. Combining Paclitaxel with Repurposed Drugs Has a Synergistic Effect on Chemoresistant HGSC Cells

Drug combination aims to achieve a therapeutic effect by taking advantage of synergism between two drugs, to help reduce the therapeutic doses and consequently minimize the associated side effects and to overcome multidrug resistance. Nevertheless, in a review from Goldin and Mantel [49], seven different definitions of synergism were described. Furthermore, another review from Greco et al. [50] mentioned 13 different methods for assessing drug synergism, demonstrating that the evaluation of synergism may have different outcomes based on the author. To date, the method proposed by Chou–Talalay for the determination of drug synergism is one of the most used in biological studies, due to its simplicity and flexibility, its quantitative definition, and its efficiency and economy. This model assumes a unified theory, based on the mass-action law-based theory, incorporating the major biochemical and biophysical equations (Henderson–Hasselbach, Hill, Michaelis–Menten, and Scatchard equations) to derive the median-effect equation that is consequently used for the determination of the combination index. Since it is supported by computer software, it has also increased its popularity due to its user-friendly interface. Another feature of this method is that it does not require the knowledge of the mechanisms of action of each drug for the determination of synergism, since the mass-action law-based determination of synergism is mechanism independent as described by Chou–Talalay [45]. This is useful, since many drugs have various mechanisms that very little is known about and others that have several mechanisms of action that make it difficult to determine which mode of action contributed to the synergy and to what extent [45].

To investigate the interaction between Paclitaxel with the previous repurposed drugs, the CI was then obtained using the Chou–Talalay method and plotted on the y-axis as a function of Fa on the x-axis to assess drug synergism. The CI indicates synergism (<1), additivity (=1) or antagonism (>1). The Fa is a parameter between 0 and 1, where 0 indicates that the drug does not affect cellular viability and 1 reveal that the drug produces a full effect on decreasing cellular viability [45,51]. The combination of Paclitaxel with Pitavastatin showed the most promising synergism with all five pairs being synergic (CI < 1) for both chemoresistant HGSC cell lines (Figure 4 and Table 1). OVCAR8 cells present a Fa value of 0.467, 0.623, 0.809, 0.850 and 0.893 for 0.25, 0.5, 1, 2 and 4 times the IC_50_ values, respectively (Figure 4A and Table 1). Similarly, OVCAR8 PTX R P cells reveal a Fa value of 0.427, 0.639, 0.759, 0.849 and 0.886 for 0.25, 0.5, 1, 2 and 4 times the IC_50_ values, respectively (Figure 4B and Table 1). Combining Paclitaxel with Metformin demonstrated synergism for two pairs in OVCAR8 cells, specifically for 0.25 and 1 times the IC_50_ values with a Fa value of 0.391 and 0.789, respectively (Figure 4A and Table 1). In addition, for OVCAR8 PTX R P cells, this combination revealed synergism for four synergic pairs for 0.25, 1, 2 and 4 times the IC_50_ values with a Fa value of 0.300, 0.734, 0.855 and 0.908, respectively (Figure 4B and Table 1). The combination of Paclitaxel with Ivermectin showed the second most promising synergism with two pairs for OVCAR8 cells, producing a Fa value of 0.913 and 0.992 for 1 and 2 times the IC_50_ values, respectively (Figure 4A and Table 1). Furthermore, this combination demonstrated synergism for OVCAR8 PTX R P cells, with a Fa value of 0.193, 0.546, 0.938 and 0.990 for 0.25, 0.5, 1 and 2 times the IC_50_ values, respectively (Figure 4B and Table 1). The combination of Paclitaxel with Itraconazole did not result in any synergism for OVCAR8 cells presenting a CI > 1 for all the pairs tested (Figure 4A and Table 1). Opposingly, for OVCAR8 PTX R P cells, this combination demonstrated synergism for four pairs with a Fa value of 0.163, 0.434., 0.652 and 0.742 for 0.5, 1, 2 and 4 times the IC_50_ values, respectively (Figure 4B and Table 1). Combining Paclitaxel with Alendronate resulted in two synergic pairs for OVCAR8 cells with a Fa value of 0.839 and 0.907 for 1 and 2 times the IC_50_ values, respectively (Figure 4A and Table 1). Likewise, for OVCAR8 PTX R P cells, this combination revealed four synergic pairs with a Fa value of 0.340, 0.825, 0.853 and 0.889 for 0.5, 1, 2 and 4 times the IC_50_ values, respectively (Figure 4B and Table 1).

Since different methods for predicting synergism can result in different outcomes, we also evaluated the drug interactions using the Bliss Independence and HSA methods (Figure 5 and Figure 6), to compare if the results corroborated the Combination Index values previously obtained by the Chou–Talalay methodology. These methods have different mathematical frameworks [52] and therefore can produce slightly different results. The Bliss independence model presumes a stochastic method in which two drugs produce their effects independently, and the expected combined effect can be assessed based on the probability of these independent events occurring [47]. The HSA method is one of the simplest synergy models and assumes that the expected combined effect is equal to the maximum of the single-drug responses at corresponding concentrations [47]. To perform these studies, we have used the SynergyFinder 2.0 Software that allows for an interactive analysis and visualization of multi-drug combination profiling data by two different synergism evaluation methods [53]. The synergy score for a drug combination is averaged over all the dose combination measurements giving a positive (synergism) or negative (antagonism) value that could be observed in 2D and 3D synergy maps dose regions, i.e., synergistic (red) and antagonistic (green) [31,54]. The Bliss Independence model revealed a positive synergy score of 9.210 and 20.025 for combining Paclitaxel with Pitavastatin, indicating additivity and synergism for OVCAR8 and OVCAR8 PTX R P cells, respectively (Figure 5A and Figure 6A). Moreover, and according to the Chou–Talalay method, the HSA model revealed that combining Paclitaxel with Pitavastatin demonstrated a stronger and positive synergy score of 16.946 and 14.399, indicating synergism for OVCAR8 and OVCAR8 PTX R P cells, respectively (Figure 5B and Figure 6B). Combining Paclitaxel with Metformin by the Bliss Independence model showed a positive synergy score of 5.378 and 10.141, demonstrating additivity and synergism for OVCAR8 and OVCAR8 PTX R P cells, respectively (Figure 5C and Figure 6C). In addition, the HSA model indicated that this combination results in a synergy score of 12.861 and 7.010, suggesting synergism and additivity for OVCAR8 and OVCAR8 PTX R P cells, respectively (Figure 5D and Figure 6D). In line with the Chou–Talalay method, the combination of Paclitaxel with Ivermectin resulted in a strong synergism, with a positive synergy score of 15.341 and 21.791 for the Bliss Independence model and 20.878 and 19.650 for the HSA model, for OVCAR8 and OVCAR8 PTX R P cells, respectively (Figure 5E,F and Figure 6E,F). The combination of Paclitaxel and Itraconazole projected by Bliss Independence and HSA models indicated synergism with a positive score of 11.367 and 20.982 and 5.889 and 18.762 for OVCAR8 and OVCAR8 PTX R P cells, respectively (Figure 5G,H and Figure 6G,H). The Bliss Independent model showed that the combination of Paclitaxel and Alendronate resulted in a positive synergy score of 4.865 and 13.108 for OVCAR8 and OVCAR8 PTX R P cells, respectively (Figure 5I and Figure 6I). The HSA model revealed that this combination results in a positive synergy score of 12.377 and 16.121, demonstrating synergism, for OVCAR8 and OVCAR8 PTX R P cells, respectively (Figure 5J and Figure 6J). Our results show that different synergy evaluation models can generate different results; however, for all the drug combinations tested, the three methods demonstrated at least one pair of concentrations with synergic behavior. Thus, we demonstrate that combining Paclitaxel with Pitavastatin and Ivermectin are the most promising combination drug pairs for both chemoresistant HGSC cell lines.

To evaluate the interaction between Paclitaxel with the repurposed drugs in HOSE6.3 (a non-tumoral cell line) we used the Chou–Talalay, Bliss Independence and HSA models. Overall, using the Chou–Talalay model, our results with the combination of Paclitaxel with Pitavastatin, Metformin, Ivermectin, Itraconazole and Alendronate in HOSE6.3 cells showed an antagonism for all five pairs being antagonistic (CI > 1) (Appendix A). Moreover, and according to the Chou–Talalay method, the Bliss independence and HSA models revealed that combining Paclitaxel with all the repurposed drugs showed a stronger and negative synergy score, indicating antagonism for HOSE6.3 cells (Figure 7).

Generally, we have found the results obtained by the Bliss and HAS reference models to be close to the ones obtained by the Chou–Talalay method, both for OVCAR 8, OVCAR 8 PTX R P, and HOSE6.3 cell lines, demonstrating that these reference models produce similar outcomes. Although the currently available reference models have been improved and demonstrated to be useful in biological studies, they still have some limitations [52]: (1) these models require ideal dose–effect curves for single drugs; (2) they are limited by the large amount of data required to make a precise synergy analysis; (3) the analysis of synergism in the clinical trials is difficult due to intense practical and ethical restrictions, which make it hard to collect sufficient data to support and evaluate drug synergism; (4) more than evaluating if a drug combination is synergistic, it is also necessary to find what dose ratio optimizes their synergy; and (5) some reference models are not adapted to drug combinations of more than two drugs.

These results also demonstrate an acceptable safety profile of the tested combination regiments in non-tumoral HOSE6.3 cells and reinforce our results regarding the repurposed drugs tested being good candidates to use in combination with Paclitaxel in HGSC patients.

## 4. Discussion

Monotherapy systems have proven to be inadequate for the treatment of chemoresistant HGSC patients. Combining two or more antineoplastic drugs is a more suitable strategy that has been demonstrating an enhanced treatment efficacy [22,55] and a synergistic result in tumor growth inhibition [56]. The goal of combining drugs is to obtain maximum efficacy using a lower drug concentration to achieve a therapeutical effect with the lowest amount of toxicity and side effects for normal cells [57]. Combinatory regiments usually include a sensitizing agent and another that take advantage of the vulnerable state and increase its cytotoxic effect [22,53]. In advanced HGSC, combinatory regimens are widely used, being the concomitant administration of Carboplatin and Paclitaxel the backbone for initial treatments [58,59,60]. The mechanisms of action of these drugs are different, enhancing the cytotoxic effect when combined [61]. Carboplatin is an alkylating agent that binds to DNA, forming reactive platinum complexes and causing DNA cross-linking, which results in modifying DNA structures and inhibits DNA synthesis. Thereby, protein synthesis and cell proliferation are blocked [62,63]. Paclitaxel is a taxane agent that targets β-tubulin, a protein responsible for stabilizing the microtubule polymers, blocking cells in phases G0/G1 and G2/M leading to tumor cells death [6]. However, when combined, the therapeutical response rate is still low, resulting in severe side effects and chemoresistance limiting maximum benefits achievement [64,65]. Hence, new models of drug combination are urgently needed to decrease the chemotherapeutic dosage, exposure time and consequentially overcome chemoresistance. Combining antineoplastic agents with repurposing drugs could be a promising approach, as this strategy increases the therapeutic efficacy by targeting multiple signaling pathways in a synergistic or, at least, additive manner [22,53,66,67].

Paclitaxel is a well-recognized antineoplastic agent used in OC management, and in a previous study, we analyzed the anti-tumor activity of this antimitotic agent in OVCAR8 and OVCAR8 PTX R P cell lines [36]. Many studies have been made in search for new drugs that can synergize with antineoplastic drugs, but few reports have been performed using repurposed drugs with Paclitaxel for HGSC management [68]. Recently, Hirst et al. demonstrated that Licofelone, an analgesic and anti-inflammatory agent, synergizes with Paclitaxel in OC by reversing chemoresistance and tumor stem-like properties [68]. Another study showed that CEP-1347 and AS602801, which targets survivin expression, sensitize OC cells to Carboplatin and Paclitaxel [69,70]. In other tumoral contexts, some reports have been made in combining Paclitaxel with different repurposing agents, e.g., Duarte et al. demonstrated that antimalarial drugs, such as Chloroquine, Artesunate and Mefloquine, could act synergistically with Paclitaxel for breast cancer therapy [31,32]. More recently, Branco et al. demonstrate that Pirfenidone, an antifibrotic drug, sensitizes non-small cell lung cancer cells to Paclitaxel [71].

Pre-clinical and retrospective studies suggest that statins, inhibitors of 3-hydroxy-3-methylglutaryl-coenzyme A reductase (HMGCR), used to treat cardiovascular diseases, can have anti-cancer activity in different tumoral contexts, including OC [72,73,74,75]. De Wolf et al. showed that Pitavastatin, a lipophilic statin with a long half-life, inhibits the growth of OC cell lines with a Carboplatin-resistant profile, suggesting a great potential to treat chemoresistant tumors [39]. In an OC setting, experimental evidence suggests that Metformin, widely used in the treatment of type 2 diabetes since it induces anti-hyperglycemia, potentiates the effectivity of chemotherapeutic agents such as platinum and taxane compounds, and reverses chemoresistance states [40,76,77,78,79,80]. As an anti-neoplastic agent, it has been demonstrated that Ivermectin, a broad-spectrum antiparasitic agent, enhances the anti-cancer efficacy of chemotherapeutic drugs and, in some cases, can reverse chemoresistance [81,82,83]. Several experimental and clinical data show promising results regarding Itraconazole, a broad-spectrum antifungal agent, revealing that combining this agent with other therapeutic drugs can be effective in several types of cancers, increasing drug efficacy [21,84,85,86] and reversing Paclitaxel chemoresistance [42,87,88,89]. Some studies demonstrate that bisphosphonates (e.g., Alendronate and Zoledronic Acid), a potent inhibitor of bone resorption used for the treatment and prevention of osteoporosis, exhibit an antineoplastic property and a capacity to delay recurrences when combined with neoplastic agents [90,91,92]. The results from previous studies reveal the anti-neoplastic potential of these drugs for cancer management.

To the best of our knowledge, this is a unique study that explores the potential synergic effect of combining Paclitaxel with Pitavastatin, Metformin, Ivermectin, Itraconazole and Alendronate in chemoresistant HGSC. Our aim was to study the potential anti-cancer activity of these five repurposed drugs in two chemoresistant HGSC cell lines (e.g., OVCAR8 (Carboplatin-resistant) and OVCAR8 PTX R P (Carboplatin and Paclitaxel-resistant)_ and to evaluate the possible synergistic effects when combined with Paclitaxel. All the drugs were screened by PB assay in a wide range of concentrations to determine their IC_50_ values and evaluate their potential as repurposed drugs in HGSC management. Our results demonstrate that all repurposed drugs used as single agent showed some antitumor activity by decreasing cellular viability in a concentration-dependent manner in both chemoresistant HGSC cell lines. All five drugs were selected for combination with Paclitaxel, and cells were treated with the concentrations of 0.25, 0.5, 1, 2 and 4 times the IC_50_ of each drug, alone and in combination with Paclitaxel, a combination model previously described by Duarte and Vale [32]. The anti-tumor effect of combining Paclitaxel with Pitavastatin, Metformin, Ivermectin, Itraconazole or Alendronate is compared to Paclitaxel alone for OVCAR8 and OVCAR8 PTX R P cells. In concomitant combination, we have shown that Pitavastatin and Ivermectin were the most promising candidates to improve Paclitaxel effectivity in both chemoresistant HGSC cell lines. Cell morphology was also analyzed after each treatment, and the phenotype of cells treated agreed with PB assay results. Next, we evaluated synergism by three different methods, e.g., Chou–Talalay, Bliss Independence and HSA models. The Chou–Talalay method is based on the median-effect equation, derived from the mass-action law principle and encompassing the Michaelis–Menten, Hill, Henderson–Hasselbach, and Scatchard equations in biochemistry and biophysics that in drug combinations provide a quantitative definition of an additive (CI = 1), synergic (CI < 1) and antagonistic (CI > 1) effects [45,51]. According to the Bliss Independence method, the two drugs produce independent effects, and the expected combination effect could be calculated based on the probability of independent events [47,53,54,93]. Furthermore, the HSA model states that the expected combination effect is the highest effect achieved by one of the two drugs (i.e., more effective drug) [31,93]. Through this synergy analysis, we showed that all tested repurposed drugs can synergistically decrease cellular viability when combined with Paclitaxel, with Pitavastatin and Ivermectin being the most promising drugs at lower concentrations for OVCAR8 and OVCAR8 PTX R P cells. Our results revealed more synergistic pairs for OVCAR8 PTX R P compared to OVCAR8 cells with almost all tested combinations resulting in synergic pairs for the lowest concentrations. Despite this, the results of single-drug treatments with repurposed drug in OVCAR8 (only Carboplatin resistant) are encouraging of the potential effect of these drugs in a platinum-resistant scenario.

Pitavastatin was the most effective repurposed drug with the lowest IC_50_ values, resulting in four and five synergic pairs when combined with Paclitaxel, for OVCAR8 and OVCAR8 PTX R P cells, respectively. Our results suggest that the dominant behavior of drug combination comes from Pitavastatin, and that the mechanism of these drug combinations may be related to this class of drug. As an anticancer agent, Statins inhibit HMGCR, leading to the blocking of cholesterol biosynthetic pathways [73,75,94]. Martirosyan et al. demonstrated that Lovastatin triggers apoptosis of OC cells as a single agent by blocking HMGCR activity and sensitizing chemoresistant cells to Doxorubicin by blocking drug efflux pumps [72]. Furthermore, results from the literature indicate that Pitavastatin is not a substrate of P-glycoprotein (P-gp), contrary to other statins such as Simvastatin, Lovastatin, and Atorvastatin [95]. We hypothesize that the synergic effect in combining Pitavastatin with Paclitaxel occurs by a similar mechanism; however, more studies are required to elucidate the mechanism behind the observed results. Concerning pharmacotherapy, the ideal drug is the one that selectively kills neoplastic cells, minimizing adverse effects in normal cells. We tested the collateral effects of these drugs in a normal-like cell line, and Pitavastatin showed to have a lack of efficacy in the reduction of cellular viability of HOSE6.3, revealing to be a non-toxic drug toward normal cells.

Concerning Metformin, many mechanisms of anti-cancer activity have been proposed, such as modulation of immunological and/or anti-inflammatory responses, inhibition of mTOR, and inhibition of the insulin signals and glucose synthesis via respiratory-chain complex I blockage [94,96,97,98,99]. Our results agree with Lengyel et al., who showed an inhibition of OC growth and an increased Paclitaxel sensitivity when using Metformin [76]. However, in our assays, the concentration of Metformin necessary to achieve an effect is higher (mM), and it will be difficult to translate these results for the clinic.

Different mechanisms can explain the anti-cancer activity of Ivermectin, including inhibition of MDR, modulation of Akt/mTOR, and Wnt/TCF signaling pathways, inactivation of PAK-1 expression, among others [41,100,101,102,103,104,105]. The results obtained with Ivermectin are in accordance with the ones obtained by Kodama et al., which showed an anti-tumoral effect of Ivermectin alone and in combination with Paclitaxel, producing a synergistic effect versus each drug alone in the OC context [81]. Moreover, Ivermectin is associated with a significantly augmented Cisplatin inhibitory effect by suppressing the phosphorylation of key molecules in the Akt/mTOR signaling pathway [82]. The anti-neoplastic effect of this antiparasitic drug has been reported to be capable of reversing the chemoresistance in colorectal, breast, and chronic myeloid leukemia cancer cells by inhibiting the EGFR/ ERK/Akt/ NF-κB pathway [83]. The combination of Paclitaxel and Ivermectin showed a significant anti-neoplastic effect in our two chemoresistant cell lines even when treated with a half IC_50_ dose of each drug.

Itraconazole targets different oncobiology mechanisms, including reversing chemoresistance mediated by P-gp, inhibiting Hedgehog, mTOR, and Wnt/β-catenin signaling pathways, and reducing angiogenesis and lymphangiogenesis [84,106,107,108,109,110]. Choi et al. demonstrated a synergistic effect of combining Itraconazole with Paclitaxel enhancing the efficacy in xenograft and patient-derived xenografts models derived from OC chemoresistant patients [42]. The available preclinical and clinical trial data indicate that Itraconazole is capable of reversing Paclitaxel chemoresistance [87,111,112]. In our study, results concerning Itraconazole did not obtain an IC_50_ value stable for being used in combination studies, make invaluable any feasible conclusions.

Bisphosphonates block farnesyl pyrophosphate synthase, located downstream of HMGCR, leading to the impairment of cholesterol biosynthesis [113]. Bisphosphonates present an anti-metastatic and anti-tumoral property when combined with chemotherapeutic agents inhibiting tumor proliferation and dissemination [91,92,114]. Experimental evidence indicates that Alendronate reduces stromal invasion, tumor burden, and ascites, suggesting an anti-tumoral effect in OC [43]. Our results are in line with the one obtained by Knight et al. where they show a direct activity of bisphosphonates (i.e., alendronate, clodronate and zoledronic acid) in five OC cell lines and tumor-derived cells; however, when combined with cytotoxic agents (Cisplatin and Paclitaxel), they do not substantially increase the activity of chemotherapeutic agents [115].

P-gp, also known as multidrug resistance protein 1 (MDR1), functions as a transmembrane efflux pump, preventing cellular uptake of a large number of structurally and functionally diverse compounds, including antineoplastic drugs [116]. P-gp overexpression is reported to be the major resistance mechanism to chemotherapeutic agents, such as Paclitaxel [117,118]. In a previous study, we demonstrated that OVCAR8 PTX R P cells presented P-gp overexpression, which allowed for the efflux of Paclitaxel outside the cells [36]. Based on literature findings, we can consider that Metformin, Ivermectin and Itraconazole may reverse chemoresistance mediated by P-gp blockage [119,120,121,122,123,124]. These drugs are substrates and modulators of P-gp and can act as chemosensitizers since they can inhibit this protein function by blocking drug efflux capacity, increase the intracellular drug accumulation, and enhance antineoplastic drug efficacy [119,120,121,122,123,124]. We hypothesize that the studied drugs could sensitize tumor cells and enhance the antineoplastic drug effect; however, more studies are needed to evaluate the anticancer mechanisms underlying different combinations.

## 5. Conclusions

Drug repurposing approaches either alone or when combined with Paclitaxel could help in reducing chemoresistance and improve treatment outcomes. Combining antineoplastic agents and repurposed drugs with independent mechanisms of action may be a promising strategy since it will suppress different chemoresistance mechanisms/pathways. Our study shows promising results concerning a potential effect of repurposed drugs as chemosensitizers compounds to enhance Paclitaxel cytotoxic effects. To the best of our knowledge, this is the first study demonstrating that Pitavastatin and Ivermectin alone and in combination with Paclitaxel are viable as a therapeutic approach that can be used on chemoresistant HGSC patients. Since both drugs are clinically available, their use in anti-cancer treatment is feasible. Although in pharmacological studies, the most important results are the effect of combining drugs at lower concentrations; deeper mechanistic studies are recommended to evaluate the anticancer mechanisms underlying these drugs and these combinatory regiments. In vitro studies using cell line models represent a simple and fast way to screen and suggest novel candidates for drug repurposing. However, since different patients present specific phenotypic characteristics, genotypic status and distinctive chemoresistance patterns, further investigations should be performed using ex vivo models, e.g., cancer cells obtained from malignant ascites drained from HGSC patients. Our results support further research that considers Pitavastatin and Ivermectin as chemosensitizing agents of Paclitaxel resistance. This strategy is particularly valuable for patients that have developed pharmacological resistance to conventional treatment and/or have been diagnosed with tumors with reduced therapeutical options, such as HGSC.

## Figures and Tables

**Figure 1 cancers-14-04357-f001:**
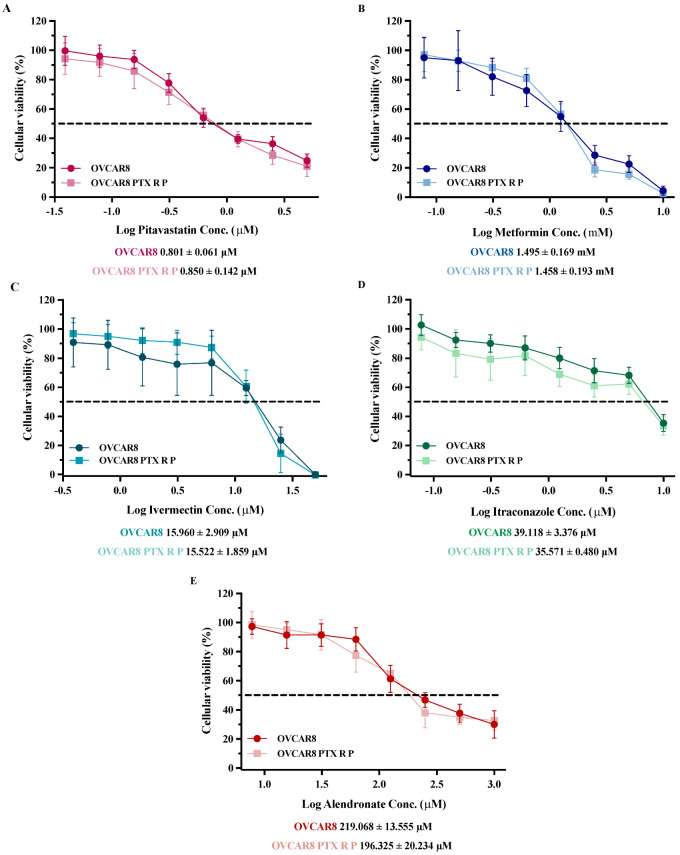
Repurposed drugs demonstrate high efficacy in reducing cellular viability of chemoresistant high-grade serous carcinoma cells. (**A**–**E**) Dose–response curves for OVCAR8 and OVCAR8 PTX R P cells obtained by Presto Blue assay after exposure to increasing concentrations of (**A**) Pitavastatin (0.04 to 5 μM), (**B**) Metformin (0.08 to 10 mM), (**C**) Ivermectin (0.39 to 50 μM), (**D**) Itraconazole (0.39 to 50 μM) and (**E**) Alendronate (7.81 to 1000 μM) for 48 h. IC_50_ values are represented by a dot line in each dose–response curve and mentioned bellow. All assays were performed in triplicate in at least three independent experiments. Data are expressed as mean ± standard deviation and plotted using GraphPad Prism Software Inc. v8 (GraphPad Software Inc., San Diego, CA, USA). Statistical analysis was performed using ordinary two-way ANOVA followed by Šidák multiple comparison test (**A**–**E**).

**Figure 2 cancers-14-04357-f002:**
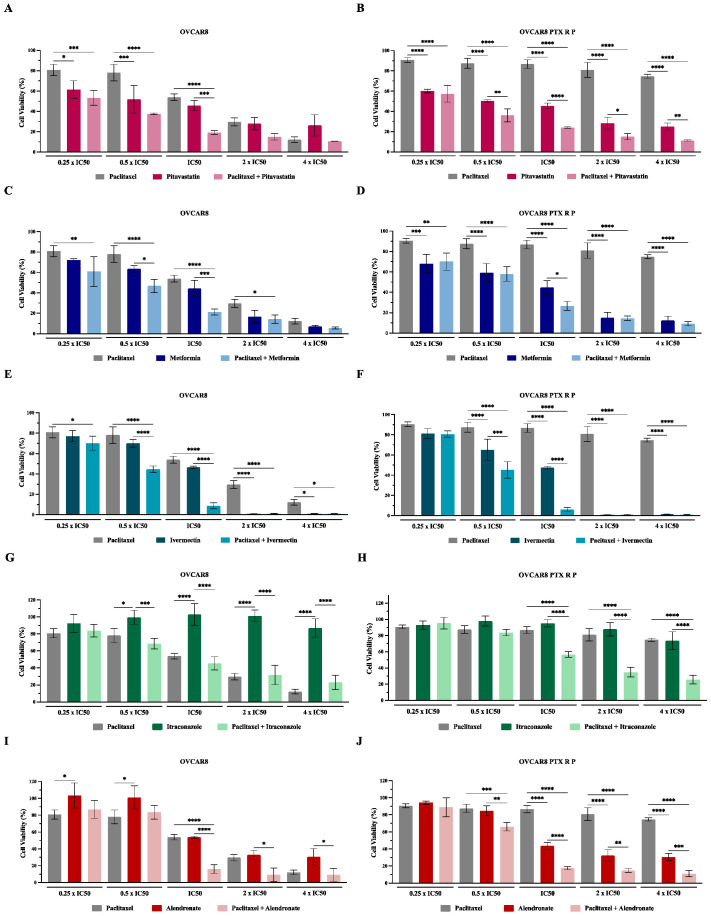
Repurposed drugs increase the efficacy of Paclitaxel in reducing cellular viability of chemoresistant high-grade serous carcinoma cells. (**A**–**J**) Bar charts showing cell viability of OVCAR8 and OVCAR8 PTX R P cells obtained by Presto Blue assay after exposure to a fixed-dose ratio that corresponds to 0.25, 0.5, 1, 2 and 4 times the individual IC_50_ values of each drug, e.g., Paclitaxel combined with (**A**,**B**) Pitavastatin, (**C**,**D**) Metformin, (**E**,**F**) Ivermectin, (**G**,**H**) Itraconazole and (**I**,**J**) Alendronate for 48 h. The combined treatment was co-administered at the same time. All assays were performed in triplicate in at least three independent experiments. Data are expressed as mean ± standard deviation and are plotted using GraphPad Prism Software Inc. v6. Statistical analysis was performed using ordinary one-way ANOVA followed by Šıdák’s multiple comparison test (**A**–**J**), and values of * < 0.05; ** < 0.001; *** <0.005; **** < 0.0001 were considered statistically significant.

**Figure 3 cancers-14-04357-f003:**
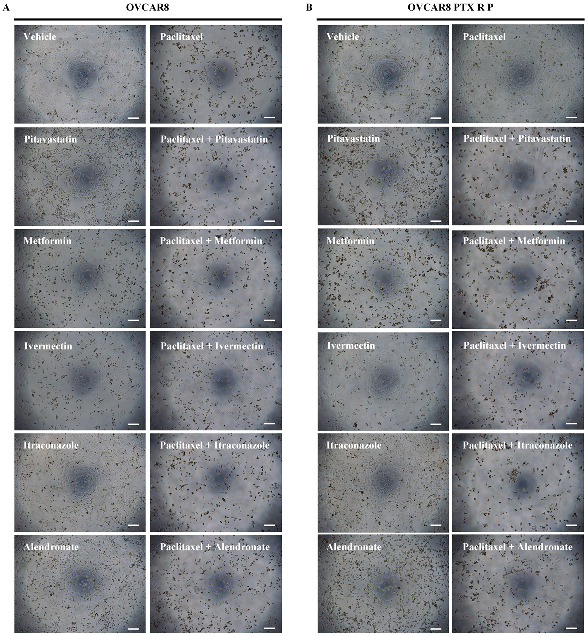
Representative microscopy images of (**A**) OVCAR8 and (**B**) OVCAR8 PTX R P cells after exposure to vehicle, Paclitaxel, Pitavastatin, Paclitaxel + Pitavastatin, Metformin, Paclitaxel + Metformin, Ivermectin, Paclitaxel + Ivermectin, Itraconazole, Paclitaxel + Itraconazole, Alendronate and Paclitaxel + Alendronate at concentration of IC_50_ values of each drug for 48 h. All assays were performed in triplicate in at least three independent experiments. Scale bar, 200 μm.

**Figure 4 cancers-14-04357-f004:**
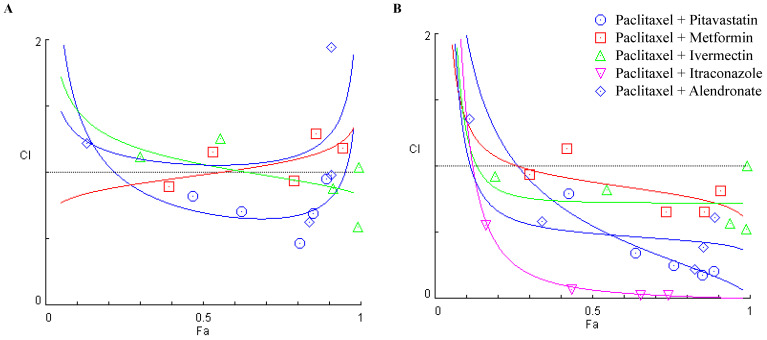
Combining Paclitaxel with repurposed drugs has a synergistic effect on chemoresistant high-grade serous carcinoma cells. (**A**,**B**) Chou–Talalay method effect level (Fa)—Combinatory Index (CI) plot showing drug synergism of (**A**) OVCAR8 and (**B**) OVCAR8 PTX R P cells, after exposure to a fixed-dose ratio that corresponds to 0.25, 0.5, 1, 2 and 4 times the individual IC_50_ values each drug, e.g., Paclitaxel combined with Pitavastatin, Metformin, Ivermectin, Itraconazole and Alendronate for 48 h. The combined treatment was co-administered at the same time. All assays were performed in triplicate in at least three independent experiments. CI was plotted on the y-axis as a function of Fa on the x-axis to evaluate drug synergism. CI: <1 (synergism), =1 (additivity) and >1 (antagonism).

**Figure 5 cancers-14-04357-f005:**
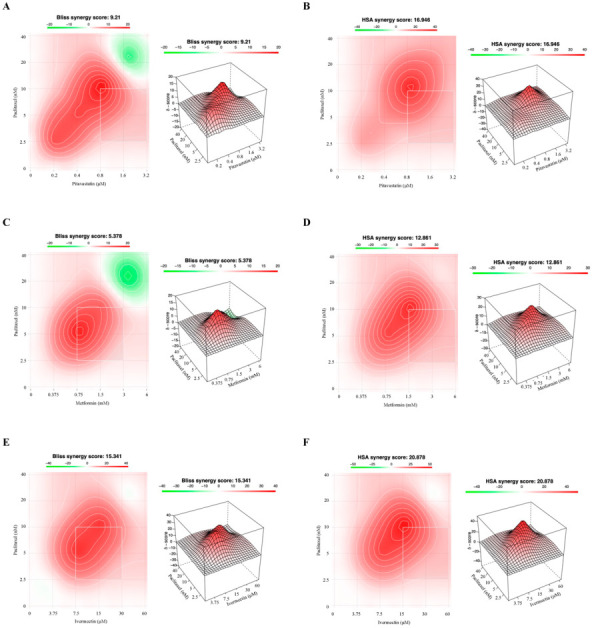
Combining Paclitaxel with repurposed drugs has a synergistic effect on OVCAR8. (**A**–**J**) Bliss Independence and High Single Agent synergy 2D and 3D plots showing drug synergism of OVCAR8 cells, after exposure to a fixed-dose ratio that corresponds to 0.25, 0.5, 1, 2 and 4 times the individual IC_50_ values for each drug, e.g., Paclitaxel combined with (**A**,**B**) Pitavastatin, (**C**,**D**) Metformin, (**E**,**F**) Ivermectin, (**G**,**H**) Itraconazole and (**I**,**J**) Alendronate for 48 h. The combined treatment was co-administered at the same time. All assays were performed in triplicate in at least three independent experiments. Synergy score: <10 (antagonism, green), =1 (additivity, white) and >10 (synergism, red).

**Figure 6 cancers-14-04357-f006:**
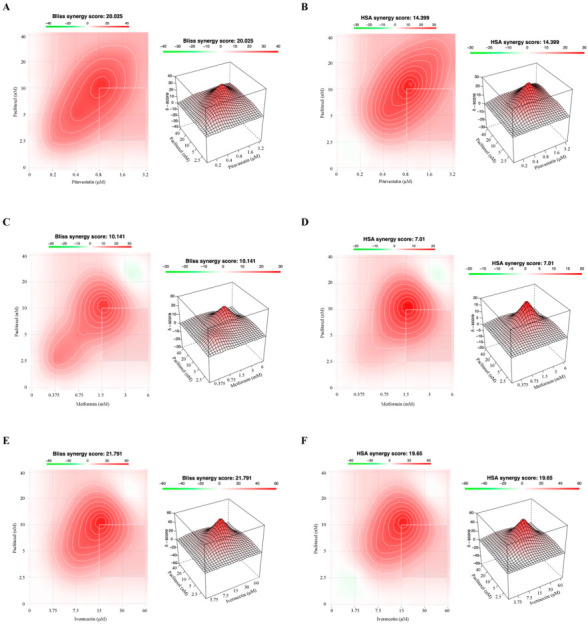
Combining Paclitaxel with repurposed drugs has a synergistic effect on OVCAR8 PTX R P. (**A**–**J**) Bliss Independence and High Single Agent synergy 2D and 3D plots showing drug synergism of OVCAR8 PTX R P cells, after exposure to a fixed-dose ratio that corresponds to 0.25, 0.5, 1, 2 and 4 times the individual IC_50_ values for each drug, e.g., Paclitaxel combined with (**A**,**B**) Pitavastatin, (**C**,**D**) Metformin, (**E**,**F**) Ivermectin, (**G**,**H**) Itraconazole and (**I**,**J**) Alendronate for 48 h. The combined treatment was co-administered at the same time. All assays were performed in triplicate in at least three independent experiments. Synergy score: <10 (antagonism, green), =1 (additivity, white) and >10 (synergism, red).

**Figure 7 cancers-14-04357-f007:**
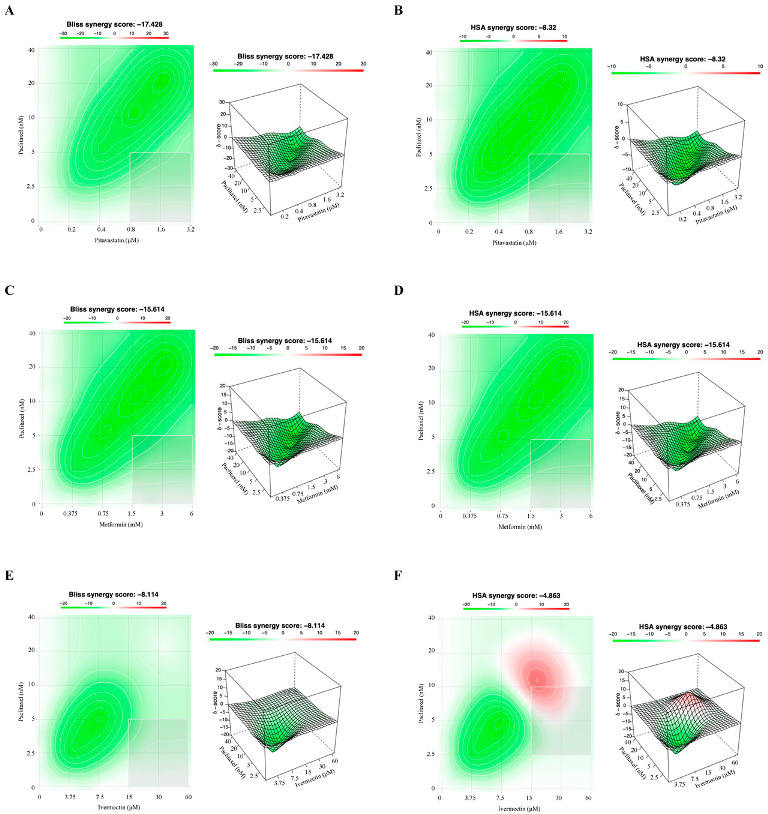
Combining Paclitaxel with repurposed drugs has an antagonistic effect on HOSE6.3 cells. (**A**–**J**) Bliss Independence and High Single Agent synergy 2D and 3D plots showing drug synergism of HOSE6.3 cells, after exposure to a fixed-dose ratio that corresponds to 0.25, 0.5, 1, 2 and 4 times the individual IC_50_ values each drug, e.g., Paclitaxel combined with (**A**,**B**) Pitavastatin, (**C**,**D**) Metformin, (**E**,**F**) Ivermectin, (**G**,**H**) Itraconazole and (**I**,**J**) Alendronate for 48 h. The combined treatment was co-administered at the same time. All assays were performed in triplicate in at least three independent experiments. Synergy score: <10 (antagonism, green), =1 (additivity, white) and >10 (synergism, red).

**Table 1 cancers-14-04357-t001:** Fractional effect (Fa) values and respective combinatory index (CI) values showing drug synergism of OVCAR8 and OVCAR8 PTX R P cells, after exposure to a fixed-dose ratio that corresponds to 0.25, 0.5, 1, 2 and 4 times the individual IC_50_ values of each drug for 48 h. The combined treatment was co-administered at the same time. All assays were performed in triplicate in at least three independent experiments. CI > 1 (antagonism), CI = 1 (additivity) and CI < 1 (synergism, bold).

				OVCAR8		OVCAR8 PTX R P	
Combination (Drug 1 + Drug 2)	Total Dose (Drug 1 + Drug 2)	Dose (Drug 1)	Dose (Drug 2)	Fa	CI Value	Fa	CI Value
**Paclitaxel (nM) + Pitavastatin (μM)**	2.7	2.5	0.2	**0.467**	**0.82421**	**0.427**	**0.79315**
	5.4	5	0.4	**0.623**	**0.70479**	**0.639**	**0.34518**
	10.8	10	0.8	**0.809**	**0.47060**	**0.759**	**0.25029**
	21.6	20	1.6	**0.850**	**0.69536**	**0.849**	**0.18051**
	43.2	40	3.2	**0.893**	**0.95039**	**0.886**	**0.20425**
**Paclitaxel (nM) + Metformin (mM)**	2.875	2.5	0.375	**0.391**	**0.89201**	**0.300**	**0.94013**
	5.75	5	0.750	0.533	1.15336	0.421	1.13628
	11.5	10	1.5	**0.789**	**0.93824**	**0.734**	**0.65835**
	23	20	3	0.859	1.29623	**0.855**	**0.65412**
	46	40	6	0.945	1.18166	**0.908**	**0.81439**
**Paclitaxel (nM) + Ivermectin (μM)**	6.25	2.5	3.75	0.300	1.12000	**0.193**	**0.92626**
	12.5	5	7.5	0.555	1.25771	**0.546**	**0.82499**
	25	10	15	**0.913**	**0.88150**	**0.938**	**0.57172**
	50	20	30	**0.992**	**0.58873**	**0.990**	**0.52306**
	100	40	60	0.994	1.04089	0.991	1.00081
**Paclitaxel (nM) + Itraconazole (μM)**	6.25	2.5	3.75	0.162	21.2198	0.048	6.07196
	12.5	5	7.5	0.315	62.8931	**0.163**	**0.55679**
	25	10	15	0.547	199.149	**0.434**	**0.06910**
	50	20	30	0.683	526.726	**0.652**	**0.02842**
	100	40	60	0.771	1307.81	**0.742**	**0.02802**
**Paclitaxel (nM) + Alendronate (μM)**	40	2.5	37.5	0.131	1.22326	0.111	1.35589
	80	5	75	0.166	2.01748	**0.340**	**0.58361**
	160	10	150	**0.839**	**0.62588**	**0.825**	**0.22371**
	320	20	300	**0.907**	**0.97858**	**0.853**	**0.38535**
	640	40	600	0.908	1.94872	**0.889**	**0.61148**

## Data Availability

The data presented in this study are available in this article and Appendix A.

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
