# Peer review of "Pitavastatin and Ivermectin Enhance the Efficacy of Paclitaxel in Chemoresistant High-Grade Serous Carcinoma"

_cancers, 2022, doi:10.3390/cancers14184357_

Round 1

Reviewer 1 Report

The overall impression of this study is that the authors convincingly provide the evidence to support their claims. It is well written and easy and interesting to read.

Author Response

We appreciate the time and effort spent evaluating our manuscript and we thank the positive feedback.

Best Regards,

Sara Ricardo

Reviewer 2 Report

Pitavastatin and Ivermectin Enhance the Efficacy of Paclitaxel in Chemoresistant High-Grade Serous Carcinoma

Nunes et al aimed to study whether the combination of Paclitaxel with repurposed drugs would lead to a therapeutic benefit by testing the cytotoxic effects of Paclitaxel alone and in combination with Pitavastatin, Metformin, Ivermectin, Itraconazole and Alendronate in chemoresistant (OVCAR8 and OVCAR8 PTX R P) cell lines and a non-tumoral (HOSE6.3) cell line.

Major comments:

1)               The study aim has clear clinical relevance. The study results have great potential for translation into ex vivo studies. The experiments performed in this study are appropriate in providing preliminary data for follow-up studies.

2)               Further clarifications are needed on the methods to better support the study analyses:

       i.          Models used in assessing drug interaction analyses:

It was mentioned in the section of materials and methods that a mutually exclusive model was used for drug interaction analyses, assuming that the drugs tested act in entirely different mechanisms. While some of the cited references and the discussion section includes information on the mechanisms of the drugs tested in the study, clearer explanation to the wider audience on why this assumption holds would help justify the use of this model for analyses.

More details on the assumptions entailed by the other additional models examined, Bliss Independence model and HSA model, would be needed to explain why these other models were reviewed, and how the related results complement with the Combination Index values in interpreting drug interaction(s).

Limitations of these models on studying drug interactions should be discussed.

     ii.          Statistical analyses:

Non-parametric tests and presentation of medians and interquartile ranges may be more appropriate than using means, SDs and ANOVA unless the normality assumption for the test is justified with the data.

   iii.          Rationale on the different ranges of drug concentrations used for the single-drug experiments should be better explained.

   iv.          It was noted in the text that “all assays were done in triplicate in at least three independent experiments.” The number(s) of experiments/samples performed for single-drug and drug combination-testing should be stated either in the text or figure legends.

Minor comments:

Typographical/grammatical issues should be fixed for the revision.

Examples for consideration:

1)     Table S2:  Periods, instead of commas, should be used for reporting percentages. Row on Ivermectin 4xIC50: cell viability percentage should not be less than 0.

2)     Page 7 of 28: second last line: “Interestingly” is a more grammatically correct term than “interesting”

3)     Page 5 of 28: Asterisks should be removed from this line of text: “p values of *<0.05; **<0.001; ***<0.0005; ****<0.0001 were considered statistically significant”

unless further explanation is given on their meaning(s) here.

4)     “All assays were done in triplicate”: The word triplicate is better to be in plural form “triplicates” given that there are multiple assays.

5)     “fixed doses ratio”: fixed-dose ratio is a more grammatically correct term.

Author Response

Response to reviewer 2 comments:

  • The study aim has clear clinical relevance. The study results have great potential for translation into ex vivo studies. The experiments performed in this study are appropriate in providing preliminary data for follow-up studies.

We appreciate the time and effort that the reviewers dedicated to providing feedback on our manuscript. We have incorporated the suggestions made by the reviewer in our manuscript which we respond with detail below.

  • Further clarifications are needed on the methods to better support the study analyses:
  1. Models used in assessing drug interaction analyses:

It was mentioned in the section of materials and methods that a mutually exclusive model was used for drug interaction analyses, assuming that the drugs tested act in entirely different mechanisms. While some of the cited references and the discussion section includes information on the mechanisms of the drugs tested in the study, clearer explanation to the wider audience on why this assumption holds would help justify the use of this model for analyses.

We thank the reviewer for the valuable feedback. We agree and have included the following paragraph in the main text (lines 324-343):

“Drug combination aims to achieve a therapeutic effect by taking advantage of synergism between two drugs, to help reduce the therapeutic doses and consequently minimize the associated side effects and to overcome multidrug resistance. Nevertheless, in a review from Goldin and Mantel [49],seven different definitions of synergism were described. Furthermore, another review from Greco et al. [50] mentioned 13 different methods for assessing drug synergism, demonstrating that the evaluation of synergism may have different outcomes based on the author. To date, the method proposed by Chou-Talalay for the determination of drug synergism is one of the most used in biological studies, due to its simplicity and flexibility, its quantitative definition, and its efficiency and economy. This model assumes a unified theory, based on the mass-action law–based theory, incorporating the major biochemical and biophysical equations (Henderson-Hasselbach, Hill, Michaelis-Menten, and Scatchard equations) to derive the median-effect equation that is consequently used for the determination of the combination index. Since it is supported by computer software, it also has increased its popularity due to its user-friendly interface. Another feature of this method is that it does not require the knowledge of the mechanisms of action of each drug for the determination of synergism, since the mass-action law–based determination of synergism is mechanism-independent as described by Chou-Talalay [45]. This is useful since many drugs have various mechanisms that very little is known about and others that have several mechanisms of action that make it difficult to determine which mode of action contributed to the synergy and to what extent [45].”

More details on the assumptions entailed by the other additional models examined, Bliss Independence model and HSA model, would be needed to explain why these other models were reviewed, and how the related results complement with the Combination Index values in interpreting drug interaction(s).

We thank the reviewer for the suggestion. We agree and have included the following explanation in the main text regarding the Bliss and HSA models (lines 419-434 and 502-513):

“Since different methods for predicting synergism can result in different outcomes, we also evaluated the drug interactions using the Bliss Independence and HSA meth-ods (Figure 5 and Figure 6), to compare if the results corroborated the Combination Index values previously obtained by the Chou-Talalay methodology. These methods have different mathematical frameworks [52] and therefore can produce slightly dif-ferent results. The Bliss independence model presumes a stochastic method in which two drugs produce their effects independently, and the expected combined effect can be assessed based on the probability of these independent events occurring [47]. The HSA method is one of the simplest synergy models and assumes that the expected combined effect is equal to the maximum of the single drug responses at corresponding concentrations [47]. To perform these studies, we have used the SynergyFinder 2.0 Software that allows an interactive analysis and visualization of multi-drug combination profiling data by two different synergism evaluation methods [53].“

“Generally, we have found the results obtained by the Bliss and HAS reference models to be very close to the ones obtained by the Chou-Talalay method, both for OVCAR 8, OVCAR 8 PTX R P, and HOSE6.3 cell lines, demonstrating that these reference mod-els produce similar outcomes.”

Limitations of these models on studying drug interactions should be discussed.

We thank the reviewer for pointing this out. We have included the following paragraph in the main text (lines 505-513):

“Although the currently available reference models have been improved and demonstrated to be useful in biological studies, they still have some limitations [52] (1) these models require ideal dose-effect curves for single drugs; (2) they are limited by the large amount of data required to make precise synergy analysis; (3) the analysis of synergism in the clinical trials is very difficult due to intense practical and ethical restrictions, which makes it hard to collect sufficient data to support and evaluate drug synergism, (4) more than evaluating if a drug combination is synergistic, it is also necessary to find what dose ratio optimizes their synergy and (5) some reference models are not adapted to drug combinations of more than two drugs..”

  1. Statistical analyses:

Non-parametric tests and presentation of medians and interquartile ranges may be more appropriate than using means, SDs and ANOVA unless the normality assumption for the test is justified with the data.

We appreciate the reviewer for this suggestion, however, almost all the articles in pharmacology that use the same genre of studies described statical analysis by means/SDS and ANOVA. We don´t find articles in the area of pharmacology that use non-parametric tests and presentation of medians and interquartile ranges.

   iii.       Rationale on the different ranges of drug concentrations used for the single-drug experiments should be better explained.

The different ranges of drug concentrations for the single-drug experiments used in this study are based on articles that use the same drugs in tumour settings. We have included the following references in the main text (lines 175-177):

“For the single drug treatment, cells were treated for 48 h with Pitavastatin (0.04 to 5 μM) [39], Metformin (0.08 to 10 mM) [40], Ivermectin (0.39 to 50 μM) [41], Itraconazole (0.39 to 50 μM) [42], and Alendronate (7.81 to 1000 μM) [43]”.

  1. It was noted in the text that “all assays were done in triplicate in at least three independent experiments.” The number(s) of experiments/samples performed for single-drug and drug combination-testing should be stated either in the text or figure legends.

Thank you, we have stated either in the text or figure legends.

Minor comments:

Typographical/grammatical issues should be fixed for the revision.

Examples for consideration:

1) Table S2:  Periods, instead of commas, should be used for reporting percentages. Row on Ivermectin 4xIC50: cell viability percentage should not be less than 0.

Thank you, we have corrected it.

2) Page 7 of 28: second last line: “Interestingly” is a more grammatically correct term than “interesting”

Thank you, we have corrected it.

3) Page 5 of 28: Asterisks should be removed from this line of text: “p values of *<0.05; **<0.001; ***<0.0005; ****<0.0001 were considered statistically significant” unless further explanation is given on their meaning(s) here.

Thank you, we have removed it.

4) “All assays were done in triplicate”: The word triplicate is better to be in plural form “triplicates” given that there are multiple assays.

Thank you, we have corrected it.

5) “fixed doses ratio”: fixed-dose ratio is a more grammatically correct term.

Thank you, we have corrected it.

We hope that we answer all your questions. Thank you very much for the positive feedback!

Best Regards,

Sara Ricardo
